# Prediction of Survival and Prognosis Migration from Gold-Standard Scores in Myelofibrosis Patients Treated with Ruxolitinib Applying the RR6 Prognostic Model in a Monocentric Real-Life Setting

**DOI:** 10.3390/jcm11247418

**Published:** 2022-12-14

**Authors:** Andrea Duminuco, Antonella Nardo, Bruno Garibaldi, Calogero Vetro, Anna Longo, Cesarina Giallongo, Francesco Di Raimondo, Giuseppe A. Palumbo

**Affiliations:** 1Postgraduate School of Hematology, University of Catania, 95123 Catania, Italy; 2Hematology Unit with BMT, A.O.U. Policlinico “G. Rodolico-San Marco”, Via S. Sofia 78, 95123 Catania, Italy; 3Dipartimento di Scienze Mediche, Chirurgiche e Tecnologie Avanzate “G.F. Ingrassia”, University of Catania, 95123 Catania, Italy

**Keywords:** RR6 prognostic score, myelofibrosis, model validation, ruxolitinib, overall survival

## Abstract

The wide use of ruxolitinib, approved for treating primary and secondary myelofibrosis (MF), has revolutionized the landscape of these diseases. This molecule can reduce spleen volume and constitutional symptoms, guaranteeing patients a better quality of life and survival or even a valid bridge to bone marrow transplantation. Despite a rapid response within the first 3 to 6 months of treatment, some patients fail to achieve a significant benefit or lose early response. After ruxolitinib failure, new drugs are available to provide an additional therapeutic option for these patients. However, the correct timing point for deciding on a therapy shift is still an open challenge. Recently, a clinical prognostic score named RR6 (Response to Ruxolitinib after 6 months) was proposed to determine survival after 6 months of treatment with ruxolitinib in patients affected by MF. We applied this model to a cohort of consecutive patients treated at our center to validate the results obtained in terms of median overall survival (mOS): for the low-risk class, mOS was not reached (as in the training cohort); for the intermediate-risk, mOS was 52 months (95% CI 39–106); for the high-risk, it was 33 (95% 8.5–59). Moreover, in addition to the other studies present in the literature, we evaluated how the new RR6 score could better identify primary MF patients at high risk, with a slight or no agreement compared to DIPSS, contrary to what occurs in secondary MF. Thus, we were able to confirm the predictive power of the RR6 model in our series, which might be of help in guiding future therapeutic choices.

## 1. Introduction

Myeloproliferative neoplasms (MPNs) are a group of hematological pathologies characterized by various degrees of overall survival (OS). Among them, myelofibrosis (MF) is undoubtedly the condition with the worst life expectancy, influenced by multiple factors. In fact, one of MF’s classic hallmarks is the extraordinary heterogeneity in presentation and the variable course of the disease. This condition can be primary (PMF) or occurs secondarily (SMF) either to polycythemia vera (PPV-MF) or to essential thrombocythemia (PET-MF). The OS may range from <2 to 20 years [1], depending on the prognostic factors at the time of diagnosis and their variations during the disease. As a matter of fact, primary versus secondary MF, somatic driver mutations, or the values in the blood count at the time of diagnosis can actively influence the prognosis [2,3,4]. Various prognostic risk classification systems have been ideated for patients with PMF. The most used score in clinical practice, based exclusively on clinical-hematological variables and applicable only at the diagnosis, is the International Prognostic Scoring System (IPSS) [5]. A dynamic prognostic model (DIPSS) was subsequently developed to evaluate patients observed during the disease course, assigning greater importance to anemia [6]. A Plus version of this score, which adds to DIPSS parameters the platelet count, the red blood cell transfusion needs, and the presence of unfavorable karyotype, has been developed [7]. This version is not used as often in clinical practice because of the lack of cytogenetics availability across all centers and the issues with obtaining accurate marrow cytogenetics in MF patients who frequently have dry taps. These two scores are more appropriate for PMF, while for SMF the MYSEC-PM dedicated score has been developed [8]. Finally, in recent years, the discovery of the crucial role of driver mutations such as JAK2, MPL, and CALR has changed our understanding of, and dealing with, MF [2]. Consequently, the availability of ruxolitinib, an inhibitor of JAK1 and JAK2, has revolutionized the MF therapeutic approach. This drug can reduce spleen dimension, ensure a rapid improvement of the symptoms, and eventually improve overall survival (OS). For these reasons, ruxolitinib is often chosen as the first-line treatment in MF [9,10,11,12,13]. However, these responses could frequently be accompanied by anemia and thrombocytopenia, especially in the early time of the treatment period [14].

Moreover, subjects who are, or become in the course of the disease, refractory or intolerant to ruxolitinib treatment pose a challenge. Several trials are still ongoing to find different therapies for this category of patients [10,15]. In this respect, the definition of the optimal timing for a treatment shift after ruxolitinib, or to undergo to hematopoietic stem cell transplant (HSCT) for eligible patients, remains an open question.

Therefore, a tool enabling the attending physician to refine an individual patient’s potential response to ruxolitinib treatment (in terms of risk of progression and overall survival) could be helpful for early identification of patients in which therapy is not giving the hoped and expected benefits and choosing the appropriate time target to shift therapy.

Maffioli et al. [16] recently published a clinical score for patients who underwent ruxolitinib treatment, intending to identify “early” predictors (after six months of ruxolitinib) of inferior survival. They set three risk factors evaluated at the baseline and after 3 and 6 months of treatment: the ruxolitinib administered dose, the palpable spleen length reduction, and the red blood cell transfusion (any quantity) requirement.

After this analysis, through the prognostic model named Response to Ruxolitinib after six months (RR6), they divided the patients into three risks categories, assigning 1 point to the presence of a single of the described variables: low (<2 points, median OS not reached), intermediate (between 2 and 4 points, median OS 61 months, 95% CI 43–80), and high (>4 points, median OS 33 months, 95% CI 21–50). Their results confirmed that the RR6 prognostic model allows for the early identification of MF patients treated with ruxolitinib and patients with defined features who might benefit from an earlier treatment shift.

While the dataset utilized by Maffioli et al. is robust and numerous (*n* = 209), it includes patients from several centers. Moreover, they based the validation of their results on a cohort of 40 patients. Recently, this model has been externally validated by Breccia et al. [17]. Thus, in the absence of other single-center studies, we decided to apply the RR6 score using a consecutive series of patients followed at our center to confirm the original model findings further.

## 2. Materials and Methods

### 2.1. Methods

First, we calculated the patients according to the DIPSS and MYSEC prognostic scores, respectively, for primary and secondary myelofibrosis at the time of ruxolitinib start and 6 months after initiation of therapy with ruxolitinib. Then, we applied the RR6 prognostic score using the calculator tool available online (https://rr6.eu/). For PMF, the choice of DIPSS over DIPSS Plus as the gold-standard score was based on the unavailability of cytogenetic testing for a large proportion of patients.

The patients were divided into three planned groups. Specifically, there were 18 patients in the low-risk class (17.5%, with a score <2 points), 53 in the intermediate (51.5%, score 2–4), and 32 in the high (31%, score >4 points). Overall survival (OS) was calculated; the median follow-up from the ruxolitinib start was 45.1 months (IQR, 23.0–87.0 months).

### 2.2. Statistical Analysis

The final data analysis was concluded on 20 September 2022. Statistical analysis was performed using R-commander software (R Foundation for Statistical Computing, Vienna, Austria). We used Kaplan-Meier estimates to analyze time-to-event data (interruption of ruxolitinib therapy due to elsewhere causes), thus comparing the three groups of subjects. Therefore, we performed a validity test (expressed with concordance index) to determine how accurately the RR6 score can stratify patients compared to the assumed gold-standard prognostic score (DIPSS for primary MF and MYSEC-PM for secondary), both calculated at the same time, i.e., the sixth month after the start of ruxolitinib therapy. To calculate the agreement, we used Cohen’s kappa, with reference to the standard intervals [18,19].

## 3. Results

### 3.1. Patient Baseline Features

This retrospective observational study aimed to validate the aforementioned RR6 score, investigating patients suffering from MF and treated at the Hematology Unit with Bone Marrow Transplantation of Policlinico “G. Rodolico-San Marco”, Catania, Italy.

A total of 103 adult patients (>18 years) with a diagnosis of primary or secondary MF were treated with ruxolitinib as the first JAK inhibitor (JAKi) therapy. Patient demographics and laboratory parameters (clinical features, complete blood count values, blood chemistry, and transfusion burden) were recorded just before ruxolitinib was started and after 3 and 6 months. As in the study of Maffioli et al., we enrolled patients with at least six months of follow-up after ruxolitinib initiation. At baseline, in all the patients, the platelet count was >40 × 109/L, the spleen was at least 5 cm below the left costal margin, and they were classified as Int-1, Int-2, or high-risk, according to the DIPSS or MYSEC-PM prognostic score.

None of the patients identified underwent allogeneic bone marrow transplantation during the period of the study. Of those patients, 62 were male (60.2%). The median age of participants was 69.4 (range: 38–83) years. According to the common prognostic scores, most patients were classified into Int-2 and high-risk classes (71.8%).

In our cohort, 57 (55.3%) patients had PMF, 46 (44.6%) were SMF, precisely 22 PET-MF, and 24 PPV-MF. All enrolled patients had bone marrow fibrosis grade ≥1, with 63 (61.2% of the total) with fibrosis grade 2 or higher. As for driver mutations, most patients were JAK2 V617-mutated (80 subjects, 77.7%). In addition, an altered karyotype was found in 19 (18.4% of the total) out of 69 patients in which cytogenetics was available. For transfusion support, 29 (28%) patients received concentrated RBC in the last three months before the start of treatment.

As suggested by guidelines, the ruxolitinib dose was chosen according to platelet level, with the most frequent starting dose of 20 mg twice daily given to 65 (63.1%) patients. The median follow-up from the MF diagnosis was 45 months [IQR 22–82].

The remaining baseline features of the enrolled patients are shown in Table 1.

### 3.2. Patient Outcomes

Out of 103 patients analyzed, two (1.9%) were lost to follow-up (respectively after 44 and 47 months of treatment, both belonging to the intermediate-risk class). At the end of the observation period, 50 patients (48.5%) died. Forty-two patients (40.8%) continued ruxolitinib therapy at the end of the observation period. In contrast, the remaining 59 (57.3%) discontinued treatment for underlying disease and treatment-related reasons (disease progression, loss of drug response, or drug intolerance-58 patients, 56.3% of total) or different unrelated reasons (one patient for post-traumatic cerebral hemorrhage in conditions of non-thrombocytopenia). One patient who discontinued ruxolitinib therapy was treated with navitoclax in a clinical study, while the others switched to the best supportive treatment.

### 3.3. Correlation by RR6 Risk Classes with DIPSS and MYSEC-PM with OS

Survival analysis with the Kaplan-Meier curve was performed by dividing the patients by the risk class calculated according to the RR6 prognostic model (Figure 1).

The median survival (mOS) was not reached for the low-risk class. For the intermediate and high-risk classes, the mOS was respectively 52 (95% CI 39–106) and 33 months (95% CI 8.5–59), with differences in OS statistically significant between the three groups (χ^2^ = 20.1, *p* = 0.00004).

We also extended the analysis and compared the groups identified with the RR6 score with the commonly used prognostic scores (DIPSS and MYSEC-PM) in the sixth month after the start of ruxolitinib therapy. In primary MF (Figure 2), patients with a high DIPSS score (four patients) were categorized as intermediate (50%) and high (50%) RR6 risk groups. Subjects with Int-2 risk (22) were mainly stratified as intermediate- or high-risk (50% and 31.8%, respectively), with four patients (18.2%) in the RR6 low-risk class. Those within the Int-1 risk class (a total of 26) were distributed between low (11.5%), intermediate (42.3%), and high (46.2%) RR6 groups. The remaining five low-risk patients were divided into low, intermediate (both 40%), and high RR6 risk (20%). For the secondary MF (Figure 3), the comparison was performed with the MYSEC-PM prognostic score. In this setting, patients with high MYSEC-PM risk (two cases) were all identified as high RR6 risk class. Most of the Int-2 patients (65.6%) fell into the RR6 intermediate category, while the rest were distributed between low-(15.6%) and high-risk (18.8%). The nine MYSEC-PM Int-1 patients were split between low (33.3%), intermediate (55.6%), and high (11.1%) classes. The low-risk were equally divided into low and intermediate classes.

Furthermore, we assessed whether the RR6 score could adequately identify and divide patients into risk classes in a similar way to the standard prognostic scores used (DIPSS and MYSEC-PM). We compared RR6 and DIPSS groups in PMF to investigate this issue, clustering them according to the reported median OS. In particular, the comparison was made between DIPSS low versus RR6 low, Int-1 versus intermediate, and Int-2 and high versus high, respectively. Based on these data, the validity indexes were 80.7% for low-risk (Cohen’s kappa = 0.05, slight agreement), 49.3% for intermediate-risk (−0.04, no agreement), and 45.6% for high-risk (−0.11, no agreement).

Similarly, in patients with SMF, we compared MYSEC-PM low and Int-1 classes versus RR6 low, Int-2 versus intermediate, and high versus high, respectively. The validity indexes were 73.3% for low-risk (k = 0.23, fair agreement), 62.2% for intermediate (k = 0.17, slight agreement), and 84.4% for high (k = 0.31, fair agreement).

## 4. Discussion

In the last ten years, significant improvements have been made in diagnosing, treating, and managing chronic myeloproliferative syndromes. The therapeutic progress was mainly due to JAKi drugs, which have revolutionized the approach to these pathologies and brought focus to myelofibrosis. While these innovations have been a turning point in managing these diseases, on the other side, early identification of patients with a non-optimal response to ruxolitinib remains an unmet need [10,15]. This aspect becomes of greater importance considering the recent marketing of new drugs potentially able to treat patients hitherto unresponsive to ruxolitinib itself [20,21,22,23,24]. In fact, there are no specific indications on the correct time point when a patient with a sub-optimal response should switch to another inhibitor, enroll in a clinical trial, or move towards an allogeneic bone marrow transplant. Therefore, the RR6 model based on the response within the first six months was of great importance for correctly identifying a subset of patients with expected reduced overall survival if kept on ruxolitinib treatment.

The RUXOREL-MF study enrolled 209 MF patients affected by primary and secondary myelofibrosis from several Italian centers and treated them with ruxolitinib as the first or subsequent line of therapy. Based on these data, four risk factors were identified: (a) dose of ruxolitinib <20 mg twice daily at baseline and months three and six, (b) palpable spleen length reduction from baseline ≤30% at months three and six, (c) red blood cell transfusion needs at month three and/or six, and (d) RBC transfusion need at all time points of observations until the sixth month. Between these, a higher ruxolitinib dose intensity (≥10 mg twice daily) is associated with better spleen response rates, especially at early time points.

We validated the RR6 model in our series of patients with MF treated with ruxolitinib in a single center, with a maximum follow-up of ten years. Our results confirmed the ability of RR6 to identify candidates for subsequent lines of treatment. Comparing our validation cohort with patients from the original study on RR6, the median of OS was not reached for patients in the low-risk class in both series. For patients with intermediate and high risk, the median OS was respectively 61 (95% CI 43–80) and 33 months (95% CI 21–50), comparable to those initially identified in the original study [16].

Concerning the assignment in the prognostic score groups (DIPSS for PMF, MYSEC-PM for SMF, and the new RR6 at 6 months from the start of therapy with ruxolitinib for all the patients), a better correlation (fair and straight) is evident in SMF patients, confirming the ability to stratify these patients through MYSEC-PM at this time point, similar to that obtained with RR6.

On the other hand, in PMF patients, there is no good correlation between intermediate- and high-risk cases, suggesting the potential usefulness of the RR6 score in these patients. A more advanced dynamic score was developed, namely DIPSS Plus [7], that evaluates the DIPSS parameters and adds to these the platelet count, the cytogenetic evaluation, and the transfusion requirements. The latter parameter is also taken into consideration in the RR score and might be one of its strengths. Moreover, the platelet count is greatly affected by ruxolitinib therapy [13] and may lead to a daily dose modification, another critical parameter needed to calculate RR6. In this respect, a comparison between RR6 and DIPSS Plus might be of greater use than the one we performed. Unfortunately, this has not been done in our series, as we lack cytogenetic analysis in most of our patients. This drawback also reflects real-world data [10], as many centers do not perform cytogenetic studies in MF patients, and the occurrence of dry tap when performing bone marrow aspiration is quite frequent in MF patients.

Furthermore, in primary MF, the RR6 score is characterized by an upgraded prognostic classification in 16 out of 57 patients and downgraded in 20. Instead, in the secondary myelofibrosis setting, 6 out of 46 patients were downgraded, and 13 were upgraded.

Finally, the analysis of the survival curves dividing into primary and secondary MF and into risk classes shows that RR6 can identify better patients at high risk compared to DIPSS (*p* = 0.2). In contrast, patients in the intermediate-risk groups have a superimposable mOS with the standard score and the recently proposed one (*p* = 0.7). Data for low-risk patients are too small to draw adequate statistical conclusions (Figure 4). On the contrary, similar survival curves are obtained with RR6 and MYSEC-PM in SMF patients (Figure 5), with the caveat that in our series, there are few patients in the high-risk groups to extrapolate definitive answers. Hence, a more extensive series is necessary to confirm these data.

## 5. Conclusions

In our series of patients with primary and secondary myelofibrosis and treated with ruxolitinib, the RR6 model can identify those patients who are less responsive to the drug and characterized by a worse prognosis, suggesting a shift in therapy towards a different JAKi or, when available, to a clinical trial and, in selected cases, to allogeneic bone marrow transplantation (HSCT). A study with a larger number of young high-risk patients who can undergo allogeneic HSCT is needed to confirm the latter issue. To reach this target, multicenter series are required, as the median age at MF diagnosis is around 68 years, and transplants are indicated for patients aged at or under 70. To summarize, this model seems to be of greater usefulness for PMF and with a better prediction power than DIPSS.

## Figures and Tables

**Figure 1 jcm-11-07418-f001:**
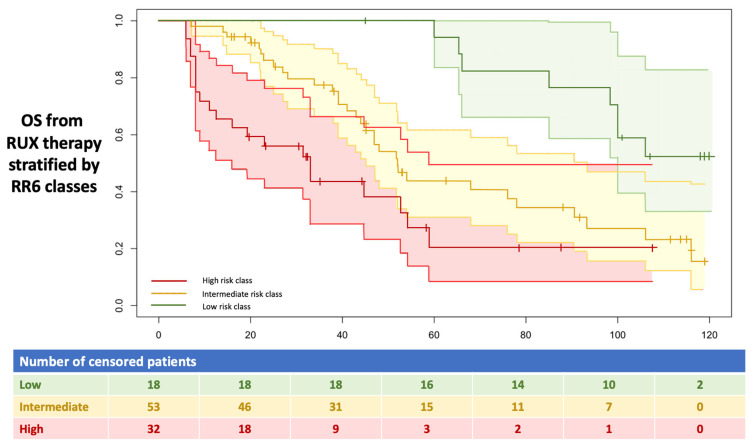
mOS in months starting from the ruxolitinib treatment. All 103 enrolled patients were stratified into three RR6 risk classes, as described in the text. RUX: ruxolitinib.

**Figure 2 jcm-11-07418-f002:**
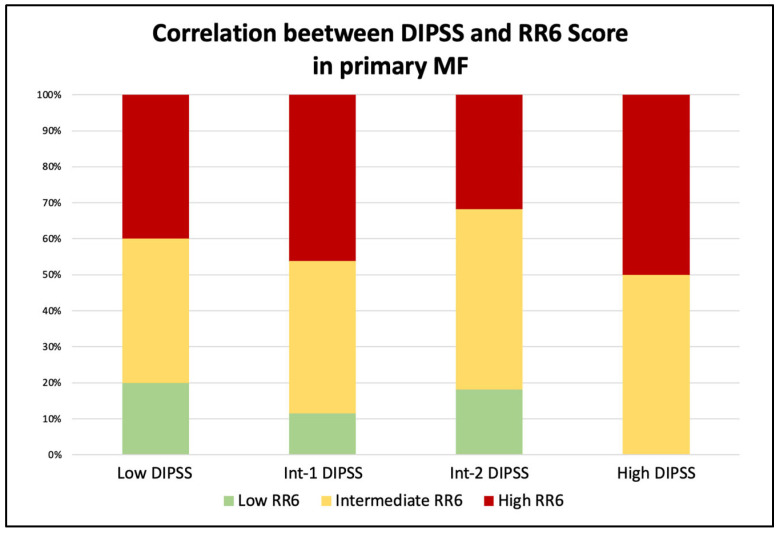
Correlation between RR6 and DIPSS risk classes in the context of primary myelofibrosis.

**Figure 3 jcm-11-07418-f003:**
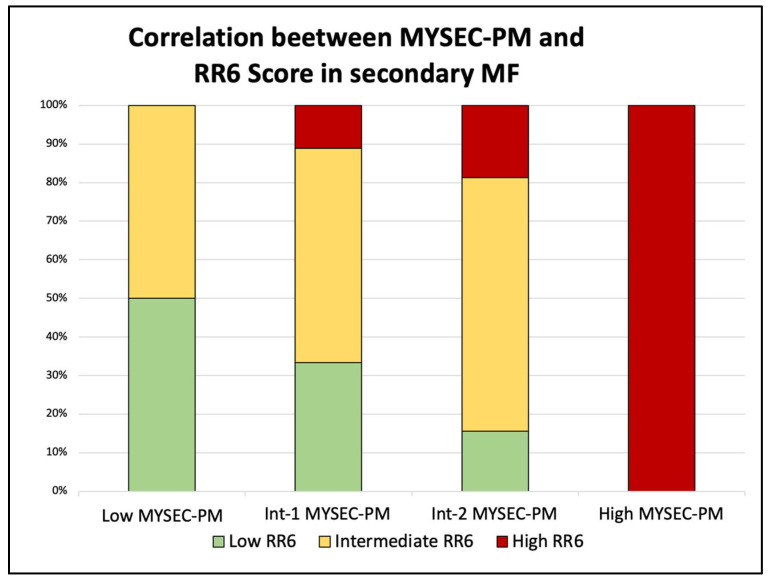
Correlation between RR6 and MYSEC-PM risk classes in the context of secondary myelofibrosis.

**Figure 4 jcm-11-07418-f004:**
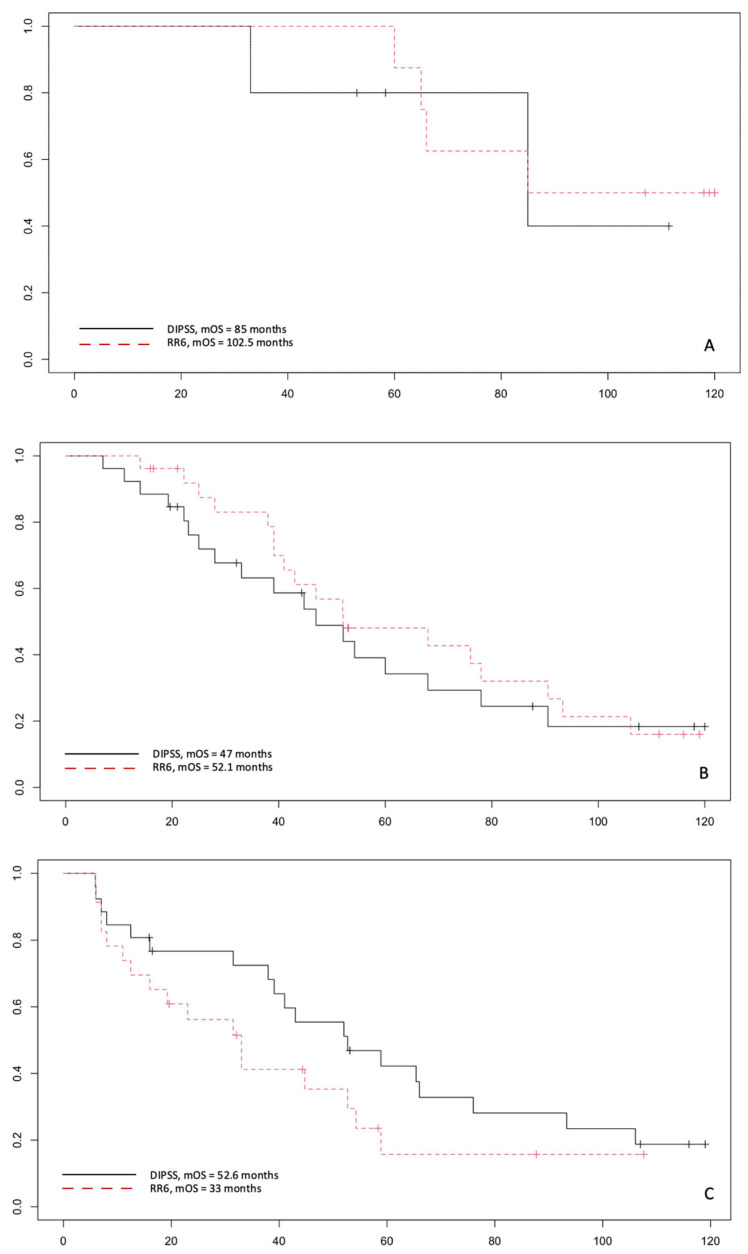
Survival curves in patients with primary myelofibrosis in risk RR6 low with DIPSS low (**A**); RR6 intermediate with DIPSS Int-1 (**B**); RR6 high with DIPSS Int-2 and high classes (**C**). NR: not reached.

**Figure 5 jcm-11-07418-f005:**
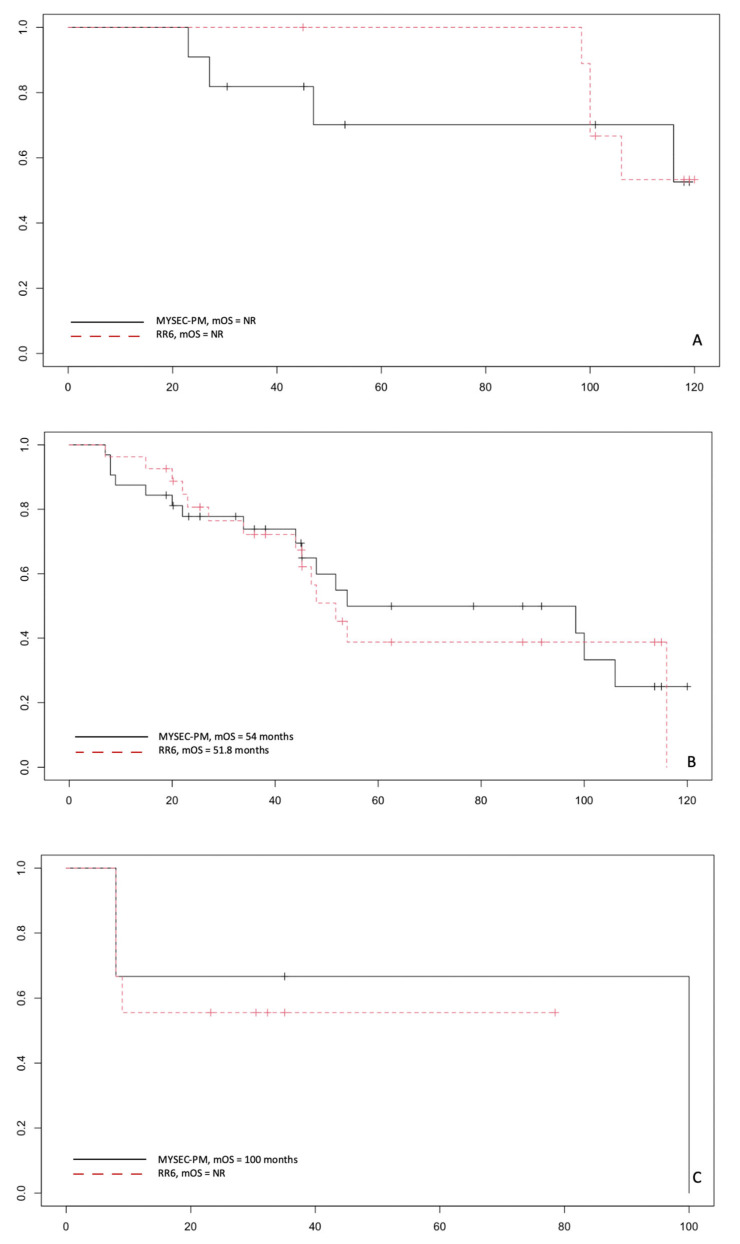
Survival curves in patients with secondary myelofibrosis in risk RR6 low with MYSEC-PM low and Int-1 (**A**); RR6 intermediate with DIPSS Int-2 (**B**); RR6 high with MYSEC-PM high classes (**C**). NR: not reached.

**Table 1 jcm-11-07418-t001:** Baseline characteristics of the 103 enrolled myelofibrotic patients. PMF: primary myelofibrosis; SMF: secondary myelofibrosis; PET: post-essential thrombocythemia; PPV: post-polycythemia vera; LCM: lower costal margin; RUX: ruxolitinib.

	At RUX Treatment Start
Median age, years (range)	*69.4 (38–83)*
Sex M/F, *n* (%)	*62 (60.2)/41 (39.8)*
PMF, *n* (%)	*57 (55.4)*
SMF, *n* (%)	*46 (44.7)*
PET-MF, *n* (%)	*22 (21.4)*
PPV-MF, *n* (%)	*24 (23.3)*
BM fibrosis grade 0/1/2/3, *n* (%)	*1 (0.9)/39 (37.9)/46 (44.7)/17 (16.5)*
Mutation status	
*JAK2-mutated, n (%)*	*80 (77.7)*
*CALR-mutated, n (%)*	*15 (14.6)*
*MPL-mutated, n (%)*	*3 (1.9)*
*‘Triple negative’, n (%)*	*6 (5.8)*
Normal/abnormal karyotype, *n* (%)	*98 (95.2)/5 (4.8)*
PMF, DIPSS LR/Int-1/Int-2/HR-n (% of PMF patients)	*0 (0)/15 (26.3)/32 (56.2)/10 (17.5)*
SMF, MYSEC-PM LR/int-1/int-2/HR-n (% of SMF patients)	*0 (0)/16 (34.8)/19 (41.3)/11 (23.9)*
Median WBC, × 10^9^/L (IQR)	*11.6 (7.8–20.3)*
Median Hb (g/dL) (IQR)	*10.8 (8.7–13)*
Median PLT × 10^9^/L (IQR)	*371 (220–554)*
Presence of 1–2% blasts in PB, *n* (%)	*4 (3.9)*
Constitutional symptoms Y/N, *n* (%)	*87 (84.5)/16 (15.5)*
Median palpable splenomegaly, cm below LCM (IQR)	*8 (4–10)*
RBC transfusions 3 months before RUX start Y/N-n (%)	*29 (28.1)/74 (71.9)*
RUX dose 5 mg BID, i.e., 10 mg total daily dose *n* (%)	*7 (6.8)*
RUX dose 10 mg BID, i.e., 20 mg total daily dose *n* (%)	*10 (9.7)*
RUX dose 15 mg BID, i.e., 30 mg total daily dose *n* (%)	*21 (2.4)*
RUX dose 20 mg BID, i.e., 40 mg total daily dose *n* (%)	*65 (63.1)*

## Data Availability

The data presented in this study are available on request from the corresponding author. The data are not publicly available due to privacy and ethical restrictions.

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
