# Peer review of "Prediction of Survival and Prognosis Migration from Gold-Standard Scores in Myelofibrosis Patients Treated with Ruxolitinib Applying the RR6 Prognostic Model in a Monocentric Real-Life Setting"

_jcm, 2022, doi:10.3390/jcm11247418_

Round 1
Reviewer 1 Report
What I like the most is how the correlation between the classical scores (DIPSS and MYSEC) with the RR6 is exposed. It is the most crucial point of this work, suggesting that response to treatment specifically in PMF could be a better tool to predict survival, but not in the case of SMF where MYSEC and RR6 show similar results (although there are few patients in this category).
This article is not written with a classical division of methods (including the statistics) and results, where I expect to find the patient´s baseline characteristics, a minor observation, which is easy to fix to be more manageable reading.
About methods: Although you consider DIPSS and MYSEC before and after six months of rux start, I need clarification on which of these points you are using to perform the correlation with RR6. If the correlation between these risk scores considered the initial point (before rux started), It would be good to know if there was a change during the first six months of treatment in these classical risk scores.
In the results, it is necessary to describe the median follow-up of patients to validate the KM.
Regarding the 3 RR6 risk groups: Are the differences in OS statistically significant between the groups?
Line 238: Figure 4: 4-C lacks the reference. All these differences in OS are statistically different or not?
Line 242: Figure 5: 5-C is not described
Line 248 “reduced prognosis”: Do you mean reduce OS?
The conclusions are consistent with the arguments, and the authors recognize that larger series are necessary.
I do not know if, with this information, you can suggest allo HSCT as an option for high-risk patients (RR6). Although High-risk RR6 patients have a median OS of 33 months, this cohort did not include patients who underwent allo-HSCT and was too small.
Author Response
Dott. Andrea Duminuco
Division of Haematology, A.O.U. “Policlinico - S. Marco”
University of Catania,
Via Santa Sofia, 78
95125 Catania, Italy
Phone number: +39 095 378 1997
Fax Number: +39 095 378 2977
e-mail address: [email protected]
Catania, 03.12.2022
To the Editorial Board of “Journal of Clinical Medicine”
Herein, we are pleased to resubmit the manuscript entitled “Prediction of survival and prognosis migration from
gold-standard scores in myelofibrosis patients treated with ruxolitinib applying the RR6 prognostic model in a
monocentric real-life setting”.
Thank you for your suggestions during the review phase.
Specifically, we enclose a table below in which we respond point by point to the notes raised during the review phase by the referees.
Sincerely,
Dr. Andrea Duminuco
REVIEWER 1
Reviewer observation |
Author response |
This article is not written with a classical division of methods (including the statistics) and results, where I expect to find the patient´s baseline characteristics, a minor observation, which is easy to fix to be more manageable reading.
|
Thank you for your observation. In the resubmitted version, we included the patient’s baseline features in the chapter of “Results” |
About methods: Although you consider DIPSS and MYSEC before and after six months of rux start, I need clarification on which of these points you are using to perform the correlation with RR6. If the correlation between these risk scores considered the initial point (before rux started), It would be good to know if there was a change during the first six months of treatment in these classical risk scores.
|
Thank you for your suggestion. The correlation between RR6 and MYSEC-PM or DIPSS is based on the 6th months reevaluation. We clarified this passage in the text. |
In the results, it is necessary to describe the median follow-up of patients to validate the KM. |
Thank you for this observation. We added in the text the median follow-up. |
Regarding the 3 RR6 risk groups: Are the differences in OS statistically significant between the groups? |
We added the statistical correlation between the three RR6 risk groups. |
Line 238: Figure 4: 4-C lacks the reference. All these differences in OS are statistically different or not?
Line 242: Figure 5: 5-C is not described |
We added the references for figures and underlined the statistically analysis |
Line 248 “reduced prognosis”: Do you mean reduce OS? |
Thank you for your observation. We corrected the terms with “worse prognosis” |
The conclusions are consistent with the arguments, and the authors recognize that larger series are necessary.
I do not know if, with this information, you can suggest allo HSCT as an option for high-risk patients (RR6). Although High-risk RR6 patients have a median OS of 33 months, this cohort did not include patients who underwent allo-HSCT and was too small. |
Thank you for your observation. We modified the conclusion, adding that our findings are able only to speculate on the possibility to send the high-risk patients to HSCT, underlying the limits of our study. Multicenter series might be needed to confirm this point as in MF the median age at diagnosis is 68 and thus few patients may be transplanted. |

Reviewer 2 Report
Major Comment
This has already been validated in a real life cohort of patients- Scalzulli et al, Blood Advances 2022. This is reference 16 that is cited and involved a cohort of 140 patients which the authors cite and discuss.
More emphasis should be made on the fact that the other studies did not look at how the RR6 differed from some patients DIPSS and MYSEC-PM scores. This should be included in the title since this is the most clinically important finding- it assists in "prognosis migration". It also needs to be mentioned in the abstract since this is the novel finding and what separates this study from previous studies.
Minor Comment
Why wasn't the DIPSS plus used instead of the DIPSS? This would be considered more of the gold standard? I would suspect it was because of a lack of cytogenetic data extending back or that the RR6 also used the DIPSS. But a comment on this would help clarify. The MYSEC-PM incorporates genetic data (CALR) so it would be interesting to see if the how the DIPSS plus performed compared to the RR6.
Figure 2 and 3 would benefit from better labeling. I am assuming the low/int/high groups are RR6 and low, int-1, int-2, and high are DIPPS and MYSEC-PM but a label on this would be nice.
Author Response
Dott. Andrea Duminuco
Division of Haematology, A.O.U. “Policlinico - S. Marco”
University of Catania,
Via Santa Sofia, 78
95125 Catania, Italy
Phone number: +39 095 378 1997
Fax Number: +39 095 378 2977
e-mail address: [email protected]
Catania, 03.12.2022
To the Editorial Board of “Journal of Clinical Medicine”
Herein, we are pleased to resubmit the manuscript entitled “Prediction of survival and prognosis migration from
gold-standard scores in myelofibrosis patients treated with ruxolitinib applying the RR6 prognostic model in a
monocentric real-life setting”.
Thank you for your suggestions during the review phase.
Specifically, we enclose a table below in which we respond point by point to the notes raised during the review phase by the referees.
Sincerely,
Dr. Andrea Duminuco
REVIEWER 2
Reviewer observation |
Author response |
More emphasis should be made on the fact that the other studies did not look at how the RR6 differed from some patients DIPSS and MYSEC-PM scores. This should be included in the title since this is the most clinically important finding- it assists in "prognosis migration". It also needs to be mentioned in the abstract since this is the novel finding and what separates this study from previous studies.
|
Thank you for your suggestion. We modified the title of the manuscript accordingly and added a short paragraph in the abstract underlying these points. |
Why wasn't the DIPSS plus used instead of the DIPSS? This would be considered more of the gold standard? I would suspect it was because of a lack of cytogenetic data extending back or that the RR6 also used the DIPSS. But a comment on this would help clarify. The MYSEC-PM incorporates genetic data (CALR) so it would be interesting to see if the how the DIPSS plus performed compared to the RR6. |
Very interesting observation. We used the DIPSS score because the cytogenetic data are unfortunately unavailable for several patients of our series, as it is a real-world study. We added an entire paragraph in the manuscript underlying this limit and discussing the point in the Discussion section. |
Figure 2 and 3 would benefit from better labeling. I am assuming the low/int/high groups are RR6 and low, int-1, int-2, and high are DIPPS and MYSEC-PM but a label on this would be nice. |
Thanks for your observation. We corrected the figures as you suggested. |

Round 2
Reviewer 2 Report
The authors have sufficiently addressed my concerns.
There are some very minor punctuation issues that could be improved.
Page 2- line 53
Would be better in my opinion to say the DIPPS Plus adds in ...... This version is not used as often in clinical practice because of the lack of cytogenetics availability across all centers and the issues with obtaining accurate marrow cytogenetics in MF patients who frequently have dry taps.
Page 9 line 281 Instead of "this minus" would say "this drawback"
Author Response
To the Editorial Board and reviewer of “Journal of Clinical Medicine”
Herein, we are pleased to resubmit the manuscript entitled “Prediction of survival and prognosis migration from gold-standard scores in myelofibrosis patients treated with ruxolitinib applying the RR6 prognostic model in a monocentric real-life setting”.
Thank you for your suggestion during this second review phase, so we modified the manuscript according to them and completed a further quick review of the English.
Dr. Duminuco